# THE POWER OF ORDER: FOOLING LLMS WITH ADVERSARIAL TABLE PERMUTATIONS

## ABSTRACT

Large Language Models (LLMs) have achieved remarkable success and are increasingly deployed in critical applications involving tabular data, such as Table Question Answering (TQA). However, their robustness to the structure of this input remains a critical, unaddressed question. This paper demonstrates that modern LLMs exhibit a significant vulnerability to the layout of tabular data. Specifically, we show that semantically-invariant permutations of rows and columns—rearrangements that do not alter the table's underlying information—are sometimes sufficient to cause incorrect or inconsistent model outputs. To systematically probe this vulnerability, we introduce **Adversarial Table Permutation (ATP)**, a novel, gradient-based attack that efficiently identifies worst-case permutations designed to maximally disrupt model performance. Our extensive experiments demonstrate that ATP significantly degrades the performance of a wide range of LLMs. This reveals a pervasive vulnerability across different model sizes and architectures, including the most recent and popular models. Our findings expose a fundamental weakness in how current LLMs process structured data, underscoring the urgent need to develop permutation-robust models for reliable, real-world applications.

## 1 INTRODUCTION

Large language models (LLMs) have demonstrated powerful reasoning capabilities, leading to significant advancements in tasks involving structured data. A key area of progress is **table question answering (TQA)**, where models interpret and extract information from tables to answer natural-language questions (Deng et al., 2024; Liu et al., 2024). The dominant paradigm for this task involves **linearizing** the table—converting its rows and columns into a serialized text format—and including it directly in the model's prompt (Zhang et al., 2024; Jiang et al., 2023; Ye et al., 2023a). This approach effectively leverages the native text-processing power of LLMs, allowing them to achieve state-of-the-art performance on some TQA benchmarks without needing specialized architectural modifications.

Despite its practical effectiveness, this linearization strategy introduces a fundamental **semantic-structural mismatch**. Tables are inherently **permutation-invariant**; their underlying relational information remains unchanged regardless of the order of their rows or columns. In stark contrast, the transformer-based architectures of LLMs are fundamentally **order-sensitive**, processing input as a strict sequence of tokens (Shi et al., 2024; Wang et al., 2025). This discrepancy creates a critical vulnerability. Because the model's understanding is tied to a superficial textual order, two tables containing identical information but presented in different layouts can elicit inconsistent and potentially incorrect outputs. This fragility undermines the reliability of LLMs in high-stakes applications and motivates a deeper investigation into their structural robustness.

Although previous work (Yang et al., 2022; Wang & Sun, 2022) demonstrated that row and column order can influence model predictions, its analysis has key limitations that restrict its applicability to modern systems. The methodology was primarily empirical, relying on random permutations to observe output changes without providing a systematic understanding of *how* specific layouts affect model reasoning. Furthermore, this research concentrated on BERT-style models for representation learning, a paradigm fundamentally different from the now-prevalent decoder-only LLM setting, where tables are processed generatively as part of an in-context prompt. This focus on older architectures offers limited guidance on the robustness of the large-scale models used in today's applications.

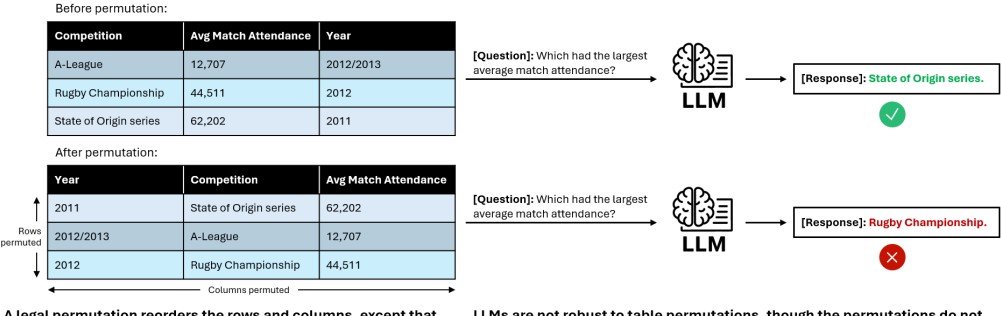

Figure 1: An illustration of the attack space for tabular inputs. A legal permutation reorders the rows and columns, except that the header row is always the first row. Such seemingly simple permutations do not change the information of the table, but are sometimes sufficient to fool modern LLMs.

In this work, we present a more rigorous investigation. We first demonstrate the susceptibility of modern LLMs to row and column permutations and then formalize this permutation sensitivity for TQA. Building on this, we introduce **Adversarial Table Permutation (ATP)**, a gradient-based attack that finds worst-case permutations by relaxing the discrete search problem into a continuous space. Across a range of instruction-tuned LLMs, ATP consistently uncovers worst-case permutations that significantly degrade prediction consistency and task performance (an example can be found in Section A.3). These adversarial layouts are transferable across model families and prompting styles and remain effective under commonly used prompting strategies. Our findings reveal a fundamental structural vulnerability in the prevailing "linearize-then-prompt" paradigm, underscoring the urgent need for more robust table reasoning techniques with LLMs.

We summarize our contributions as follows:

- We formalize the vulnerability of modern LLMs to permutations in tabular inputs, demonstrating that even random shuffling of rows and columns is sometimes sufficient to degrade model performance.
- To systematically expose this weakness, we propose the **Adversarial Table Permutation (ATP)** attack, a novel, gradient-based method that efficiently finds the worst-case permutations that cause a target model to fail. Our method is quite general and serves as a module that works for any open source LLM that takes embeddings, position ids, and attention masks as input.
- We conduct extensive experiments showing that ATP successfully degrades the performance of a wide range of LLMs, including those of various sizes and architectures. These findings reveal a critical design flaw in how current models handle structured data, underscoring the need for more robust architectures for real-world tabular applications.

## 2 PROBLEM SETTING

### 2.1 ATTACK SPACE FOR TABULAR INPUT: ROW AND COLUMN PERMUTATIONS

Given tabular data as input, a robust model is expected to produce outputs that are **invariant** to row and column permutations that preserve the table's semantic meaning. Specifically, given a table with $n + 1$ rows and $m$ columns where the first is a header row, one can arbitrarily permute the remaining $n$ data rows and all $m$ columns without changing the underlying relational information. An example of such a semantically equivalent permutation can be seen in Figure 1, where the original and permuted tables contain the exact same information.

Formally, we define this attack space in the context of Table Question Answering (TQA) tasks, where we have i.i.d. samples of an input table $\mathbf{T}$, a question $\mathbf{Q}$, and an answer $\mathbf{A}$ from a given data distribution. Here, $\mathbf{Q}$ and $\mathbf{A}$ are both sequences of words, while $\mathbf{T}$ is represented as an $(n + 1) \times m$ matrix where each cell contains a sequence of words. Let $\Pi_k$ be the set of all $k \times k$ permutation matrices. Then the attack space for the input table $\mathbf{T}$ is defined as:

$$\boldsymbol{P}_r\mathbf{T}\boldsymbol{P}_c, \quad \text{s.t., } \boldsymbol{P}_r \in \hat{\Pi}_{n+1}, \boldsymbol{P}_c \in \Pi_m, \tag{1}$$

$$\text{where } \hat{\Pi}_{n+1} := \big\{\boldsymbol{P} \in \Pi_{n+1} : \boldsymbol{P}_{[0,0]=1}\big\}. \tag{2}$$

In Equation (1), the matrix $\boldsymbol{P}_r$ permutes the rows of $\mathbf{T}$ and $\boldsymbol{P}_c$ permutes the columns. The constraint $\boldsymbol{P}_{[0,0]} = 1$ in the definition of $\hat{\Pi}_{n+1}$ ensures that the header row is always the first row. Given this formulation, we next discuss the key research problems this work aims to address.

## 2.2 Are LLMs Robust Against Table Permutations?

The key motivation of this work is to investigate to what extent current LLMs are robust against row and column permutations of the input table. This can be decomposed into three key research questions: (i) Are current LLMs robust to random table permutations? (ii) How to generate the worst-case table permutation to fool a LLM? (iii) To what extent current LLMs are robust to the worst-case table permutations? We will first formalize (i) and (ii) in the rest of this section and then address (ii) in Section 3 by proposing a novel attack method, and finally answer (i) and (iii) in our experiments in Section 5.

Consider a LLM that parametrizes a probability mass function over the natural language space, as $P_{\mathrm{model}}(\cdot)$. To employ the LLM for TQA tasks, we generate the model response by sampling from the parameterized conditional distribution, as $\hat{\mathbf{A}} \sim P_{\mathrm{model}}(\cdot \mid \mathbf{T}, \mathbf{Q})$. Then we evaluate to what extent $\hat{\mathbf{A}}$ is semantically aligned with the ground truth $\mathbf{A}$, by some evaluation metrics $\mathcal{M}(\mathbf{A}, \hat{\mathbf{A}})$ (the bigger the better alignment).

Thus, the research problem (i) can be formulated as the calculation of the following

$$\mathbb{E}_{\hat{\mathbf{A}} \sim P_{\mathrm{model}}(\cdot \mid \boldsymbol{P}_r \mathbf{T} \boldsymbol{P}_c, \mathbf{Q}),\ \boldsymbol{P}_r \sim \mathcal{U}_r,\ \boldsymbol{P}_c \sim \mathcal{U}_c} \mathcal{M}(\mathbf{A}, \hat{\mathbf{A}}), \tag{3}$$

where $\mathcal{U}_r$ and $\mathcal{U}_c$ are uniform distribution over $\hat{\Pi}_{n+1}$ and $\Pi_m$, respectively.

As for (ii), it can be formulated as finding the worst case combination of row and column permutations $(\boldsymbol{P}_r^*, \boldsymbol{P}_c^*)$ to fool a victim model $P_{\mathrm{model}}$, as,

$$(\boldsymbol{P}_r^*, \boldsymbol{P}_c^*) = \underset{\boldsymbol{P}_r \in \hat{\Pi}_{n+1}, \boldsymbol{P}_c \in \Pi_m}{\arg\min} \mathbb{E}_{\hat{\mathbf{A}} \sim P_{\mathrm{model}}(\cdot \mid \boldsymbol{P}_r \mathbf{T} \boldsymbol{P}_c, \mathbf{Q})} \mathcal{M}(\mathbf{A}, \hat{\mathbf{A}}), \tag{4}$$

and then evaluate the performance under such worst case permutations, as

$$\mathbb{E}_{\hat{\mathbf{A}}^* \sim P_{\mathrm{model}}(\cdot \mid \boldsymbol{P}_r^* \mathbf{T} \boldsymbol{P}_c^*, \mathbf{Q})} \mathcal{M}(\mathbf{A}, \hat{\mathbf{A}}^*). \tag{5}$$

By Equation (3), it is straightforward to evaluate the robustness of a LLM against random table permutations. As a contrast, the optimization problem in Equation (4) is highly nontrivial. Solving the combinatorial optimization problem in Equation (4) directly in the permutation space $\hat{\Pi}_{n+1}$ and $\Pi_m$ is NP-hard. The computation complexity grows exponentially with the shape of the input table. For example, when $n = m = 8$, there are around $1.6 \times 10^9$ different kinds of combinations of row and column permutations, and this number increases to $1.3 \times 10^{11}$ when $n$ and $m$ are increased by only 1. Therefore, it is crucial to have a more effective way to find the worst case permutation, and we propose our novel method in what follows.

## 3 Adversarial Table Permutation (ATP) Attack

The core challenge in finding the worst-case permutation, as formulated in Equation (4), is that it requires optimizing over a vast and discrete space of permutation matrices, which is computationally intractable for tables of non-trivial size. To overcome this hurdle, our proposed Adversarial Table Permutation (ATP) attack reframes this discrete problem into a continuous one that can be solved efficiently with gradient-based methods. This is achieved through two key relaxations, which are detailed below. An illustration of the overall process can be found in Figure 2.

### 3.1 Relaxing the Discrete Problem into a Differentiable One

**From a Non-Differentiable Metric to a Differentiable Loss.** Our first step is to transform the optimization objective into a continuous, differentiable form. Specifically, the evaluation metric $\mathcal{M}(\mathbf{A}, \hat{\mathbf{A}})$ is often non-differentiable and requires sampling model outputs $\hat{\mathbf{A}}$, leading to noisy and unstable gradients. We replace this objective with the maximization of the standard cross-entropy

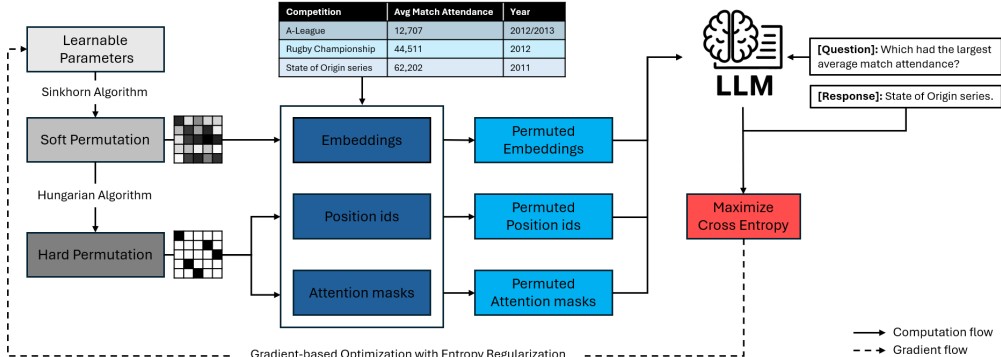

Figure 2: An illustration of the whole procedure of ATP attack, where soft permutations (doubly stochastic matrices) and hard permutations are parameterized by learnable parameters, and then used to permute embeddings of the input table and position ids and attention masks of the table, respectively. The permuted input is fed to LLM together with the question and ground truth response to calculate cross entropy loss. Finally the gradient of the loss is employed to update the learnable parameters for maximizing the cross entropy and thus fooling the victim model.

loss $\mathcal{L}_{\text{CE}}$ of the ground-truth answer $\mathbf{A}$. This provides a differentiable proxy that directly measures the alignment of model output with the ground truth answer. The optimization problem thus becomes

$$(\boldsymbol{P}_r^*, \boldsymbol{P}_c^*) = \underset{\boldsymbol{P}_r \in \hat{\Pi}_{n+1}, \boldsymbol{P}_c \in \Pi_m}{\arg\max} \mathcal{L}_{\text{CE}}(P_{\text{model}}, \boldsymbol{P}_r \mathbf{T} \boldsymbol{P}_c, \mathbf{Q}, \mathbf{A}), \tag{6}$$

where the cross-entropy loss is as follows:

$$\mathcal{L}_{\text{CE}}(\cdot) = -\sum_{t=0}^{|\mathbf{A}|-1} \log P_{\text{model}}(\mathbf{A}_{[t]} | \boldsymbol{P}_r \mathbf{T} \boldsymbol{P}_c, \mathbf{Q}, \mathbf{A}_{[:t]}). \tag{7}$$

Here $\mathcal{L}_{\text{CE}}$ is the sum of the cross-entropy for each token $\mathbf{A}_{[t]}$ conditioned on the correct context. It provides a stable, gradient-friendly objective without the need for sampling.

**From Permutation Matrices to Doubly Stochastic Matrices.** While the objective is now differentiable, the search space of permutation matrices ($\hat{\Pi}_{n+1}$ and $\Pi_m$) remains discrete. To create a continuous search space, we perform a convex relaxation to relax the set of permutation matrices to its convex hull. The reasons to consider convex hull lies in that the convex hull is the minimal convex and continuous superset of the original discrete space, which facilitates efficient optimization.

**Theorem 1** (Birkhoff-von Neumann (Birkhoff, 1946)). *The convex hull of the set of $n \times n$ permutation matrices, $\Pi_n$, is the set of $n \times n$ doubly stochastic matrices, $\mathbb{D}_n$.*

By Theorem 1, the convex hull of $\Pi_m$ is the set of all $m \times m$ doubly stochastic matrices, $\mathbb{D}_m$. As for relaxing $\hat{\Pi}_{n+1}$, we further define $\hat{\mathbb{D}}_{n+1} = \{\boldsymbol{D} \in \mathbb{D}_{n+1} : \boldsymbol{D}_{[0,0]} = 1\}$, the set of all $n \times n$ doubly stochastic matrices whose upper-left entry is always 1. As such we can now optimize over the continuous and convex sets of doubly stochastic matrices $\hat{\mathbb{D}}_{n+1}$ and $\mathbb{D}_m$.

Specifically, we first parametrize two unconstrained real matrices $\boldsymbol{\theta}_r \in \mathbb{R}^{n \times n}$ and $\boldsymbol{\theta}_c \in \mathbb{R}^{m \times m}$, and then transform they to two "soft" permutation matrices $\boldsymbol{D}_r$ and $\boldsymbol{D}_c$, by leveraging the differentiable log-Sinkhorn algorithm (Sinkhorn, 1964; Adams & Zemel, 2011; Mena et al., 2018), as

$$\boldsymbol{S}^0(\boldsymbol{\theta}) = \boldsymbol{\theta}, \ \boldsymbol{S}^{i+1}(\boldsymbol{\theta}) = \boldsymbol{N}_c(\boldsymbol{N}_r(\boldsymbol{S}^i(\boldsymbol{\theta}))), \ \boldsymbol{S}(\boldsymbol{\theta}) = \exp(\lim_{i \to \infty} \boldsymbol{S}^i(\boldsymbol{\theta})), \tag{8}$$

where $\boldsymbol{N}_r$ and $\boldsymbol{N}_c$ are row normalization and column normalization, respectively, as

$$\boldsymbol{N}_r(\boldsymbol{\theta})_{[i,:]} = \boldsymbol{\theta}_{[i,:]} - \sum_j \boldsymbol{\theta}_{[i,j]}, \quad \boldsymbol{N}_c(\boldsymbol{\theta})_{[:,j]} = \boldsymbol{\theta}_{[:,j]} - \sum_i \boldsymbol{\theta}_{[i,j]}. \tag{9}$$

By the theorem in Sinkhorn (1964), we have $\boldsymbol{S}(\boldsymbol{\theta}_r) \in \mathbb{D}_n$ and $\boldsymbol{S}(\boldsymbol{\theta}_c) \in \mathbb{D}_m$, and then we define the soft permutation matrices $\boldsymbol{D}_r$ and $\boldsymbol{D}_c$, as

$$\boldsymbol{D}_{r[1:,1:]} = \boldsymbol{S}(\boldsymbol{\theta}_r), \ \boldsymbol{D}_{r[0,0]} = 1, \ \boldsymbol{D}_{r[0,1:]} = \boldsymbol{D}_{r[1:,0]} = 0, \ \text{and } \boldsymbol{D}_c = \boldsymbol{S}(\boldsymbol{\theta}_c), \tag{10}$$

where $\boldsymbol{D}_r$ is designed to ensure the header row remains fixed, and thus we have $\boldsymbol{D}_r \in \hat{\mathbb{D}}_{n+1}$ and $\boldsymbol{D}_c \in \mathbb{D}_m$. This allows us to search for the optimal permutation in a continuous space using gradients with respect to $\boldsymbol{\theta}_r$ and $\boldsymbol{\theta}_c$.

## 3.2 Projecting Back to the Permutation Space

The relaxation in Section 3.1 allows for gradient-based optimization, but it introduces a new challenge: the optimized matrices $\boldsymbol{D}_r$ and $\boldsymbol{D}_c$ are "soft" permutations, not the hard, discrete permutations required for a valid attack. Furthermore, an LLM's input for table $\mathbf{T}$ consists of multiple components: continuous token embeddings $\mathbf{T}^{\text{emb}}$ and discrete position ids $\mathbf{T}^{\text{pos}}$ and attention masks $\mathbf{T}^{\text{att}}$. While soft permutations can be applied to continuous embeddings, they cannot be applied to discrete inputs.

To address this, we apply the permutations differently based on the input type. We use the soft doubly stochastic matrices for the embeddings and project them back to the nearest permutation matrices for the discrete components. Such a projection is captured by a maximum weight matching problem that can be solved by the Hungarian algorithm (Kuhn, 1955; 1956), as

$$\text{Proj}_n(\boldsymbol{D}) = \underset{\boldsymbol{P} \in \Pi_n}{\arg\max} \langle \boldsymbol{P}, \boldsymbol{D} \rangle_F, \boldsymbol{D} \in \mathbb{D}_n, \tag{11}$$

where $\langle \cdot, \cdot \rangle_F$ is the Frobenius inner product and the subscript $n$ in $\text{Proj}_n(\cdot)$ is dropped when the context is clear. The input table to the model is thus permuted in a hybrid mode, as

$$P_{\text{model}}(\cdot \,|\, \boldsymbol{D}_r \mathbf{T}^{\text{emb}} \boldsymbol{D}_c, \ \text{Proj}(\boldsymbol{D}_r) \mathbf{T}^{\text{pos}} \text{Proj}(\boldsymbol{D}_c), \ \text{Proj}(\boldsymbol{D}_r) \mathbf{T}^{\text{att}} \text{Proj}(\boldsymbol{D}_c), \ \mathbf{Q}). \tag{12}$$

This hybrid approach allows us to maintain a differentiable optimization pipeline while ensuring the final generated attack is valid and correctly manipulates all aspects of the model's input.

## 3.3 Regularization: Information Entropy-based Over Temperature-based

Another key challenge in our relaxed optimization is to ensure that the resulting doubly stochastic matrices, $\boldsymbol{D}_r$ and $\boldsymbol{D}_c$, are close to actual permutation matrices. Without this constraint, the soft permutations during optimization could converge to solutions far from any single permutation (e.g., a uniform matrix), creating a significant gap between the loss measured during optimization and the attack's true effectiveness. To this end, we must encourage the soft matrices to be "sharp" and structurally similar to a hard permutation.

A classic strategy is to follow Gumbel-Softmax (Jang et al., 2016; Maddison et al., 2016) to introduce a temperature parameter $\tau$ into the Sinkhorn algorithm, as $\boldsymbol{S}(\boldsymbol{\theta}/\tau)$. Lowering $\tau$ is analogous to pushing the optimization towards low-entropy solutions and thus closer to a hard permutation. However, in practice, we found that to encourage $\boldsymbol{S}(\boldsymbol{\theta}/\tau)$ to be sufficiently close to a hard permutation, it relies on a pretty small temperature $\tau$ (e.g., $\tau \leq 0.05$); such a small $\tau$ introduces significant computational instability. This issue is exacerbated when using low-precision floating-point formats like float16 or bfloat16, which are however very common for LLMs, especially in memory-constrained scenarios.

To circumvent this instability while still achieving the same goal, we incorporate the information entropy $\mathcal{H}(\cdot)$ as an explicit regularization term in our final objective, defined as follows:

$$\mathcal{H}(\boldsymbol{D}) = -\sum_{i=1}^{n} \sum_{j=1}^{n} \boldsymbol{D}_{ij} \log(\boldsymbol{D}_{ij}), \tag{13}$$

where the input $\boldsymbol{D}$ is a $n \times n$ doubly stochastic matrix. This approach, partly inspired by Dong et al. (2021a), directly encourages the soft matrices to be close to permutation matrices without the numerical issues associated with a small temperature $\tau$. We also empirically validate this point by our ablation study in Section 5.3.

## 3.4 Final Objective

Thus, by adding the entropy term to the optimization objective, combining the differentiable loss and the hybrid permutation strategy, our final optimization objective is to find the parameters $(\boldsymbol{\theta}_r^*, \boldsymbol{\theta}_c^*)$

---

**Algorithm 1** Adversarial Table Permutation (ATP) Attack

---

1: **Input:** $P_{\text{model}}$, $\mathbf{T}$, $\mathbf{Q}$, $\mathbf{A}$, $N_{\text{attack}}$, $\lambda_1$, $\lambda_2$;
2: **Output:** Worst case row permutation $\boldsymbol{P}_r^*$ and column permutation $\boldsymbol{P}_c^*$;
3: Initialize $\boldsymbol{\theta}_r \in \mathbb{R}^{n \times n}$ and $\boldsymbol{\theta}_c \in \mathbb{R}^{m \times m}$ and get $\mathbf{T}^{\text{emb}}, \mathbf{T}^{\text{pos}}, \mathbf{T}^{\text{att}}$;
4: **for** $i = 1$ to $N_{\text{attack}}$ **do**
5:     Calculate $\boldsymbol{S}(\boldsymbol{\theta}_r)$ and $\boldsymbol{S}(\boldsymbol{\theta}_c)$ by Equation (8);
6:     Calculate $\boldsymbol{D}_r$ and $\boldsymbol{D}_c$ by Equation (10) and $\mathcal{H}(\boldsymbol{D}_r)$ and $\mathcal{H}(\boldsymbol{D}_c)$ by Equation (13) ;
7:     Calculate $\text{Proj}(\boldsymbol{D}_r)$ and $\text{Proj}(\boldsymbol{D}_c)$ by the Hungarian algorithm (Kuhn, 1955);
8:     Calculate $\mathcal{L}_{\text{CE}}(P_{\text{model}}, \boldsymbol{D}_r\mathbf{T}^{\text{emb}}\boldsymbol{D}_c, \text{Proj}(\boldsymbol{D}_r)\mathbf{T}^{\text{pos}}\text{Proj}(\boldsymbol{D}_c), \text{Proj}(\boldsymbol{D}_r)\mathbf{T}^{\text{att}}\text{Proj}(\boldsymbol{D}_c), \mathbf{Q}, \mathbf{A})+$
        $\lambda_1\mathcal{H}(\boldsymbol{D}_r) + \lambda_2\mathcal{H}(\boldsymbol{D}_c)$;
9:     Calculate the gradient of $\mathcal{L}_{\text{CE}}$ with respect to $\boldsymbol{\theta}_r$ and $\boldsymbol{\theta}_c$;
10:    Update $\boldsymbol{\theta}_r$ and $\boldsymbol{\theta}_c$ by Adam (Kingma, 2014);
11: Let $\boldsymbol{\theta}_r^*, \boldsymbol{\theta}_c^* = \boldsymbol{\theta}_r, \boldsymbol{\theta}_c$, calculate $\boldsymbol{D}_r^*, \boldsymbol{D}_c^*$ given $\boldsymbol{\theta}_r^*, \boldsymbol{\theta}_c^*$ by Equation (10);
12: Calculate $\boldsymbol{P}_r^* = \text{Proj}(\boldsymbol{D}_r^*)$ and $\boldsymbol{P}_c^* = \text{Proj}(\boldsymbol{D}_c^*)$
13: **return** $\boldsymbol{P}_r^*, \boldsymbol{P}_c^*$

---

that maximizes the weighted sum of cross-entropy loss and entropy regularization terms:

$$(\boldsymbol{\theta}_r^*, \boldsymbol{\theta}_c^*) = \underset{\boldsymbol{\theta}_r \in \mathbb{R}^{n \times n}, \boldsymbol{\theta}_c \in \mathbb{R}^{m \times m}}{\arg\max} \mathcal{L}_{\text{CE}}(P_{\text{model}}, \boldsymbol{D}_r\mathbf{T}^{\text{emb}}\boldsymbol{D}_c, \text{Proj}(\boldsymbol{D}_r)\mathbf{T}^{\text{pos}}\text{Proj}(\boldsymbol{D}_c), \quad (14)$$

$$\text{Proj}(\boldsymbol{D}_r)\mathbf{T}^{\text{att}}\text{Proj}(\boldsymbol{D}_c), \mathbf{Q}, \mathbf{A}) + \lambda_1\mathcal{H}(\boldsymbol{D}_r) + \lambda_2\mathcal{H}(\boldsymbol{D}_c), \quad (15)$$

where $\lambda_1$ and $\lambda_2$ are hyper-parameters controlling the weight of the entropy terms.

Once the solution $(\boldsymbol{\theta}_r^*, \boldsymbol{\theta}_c^*)$ are found, the resulting optimal soft matrices $(\boldsymbol{D}_r^*, \boldsymbol{D}_c^*)$ are projected via $\text{Proj}(\cdot)$ to obtain the final adversarial permutation matrices $(\boldsymbol{P}_r^*, \boldsymbol{P}_c^*)$ used to attack the LLM. In our implementation the optimization in Equation (14) is solved by Adam (Kingma, 2014), where the number of iterations $N_{\text{attack}}$ serves as a hyper-parameter (ablation study on $N_{\text{attack}}$ in Section 5.3). We also summarize the whole algorithm procedure of the proposed ATP attack in Algorithm 1.

## 4 RELATED WORK

**Adversarial Attacks and Robustness for LLM.**    LLMs remain vulnerable to adversarial manipulations that bypass predefined safety mechanisms and induce harmful or undesired outputs (He et al., 2024; Qi et al., 2024; Hsiung et al., 2025). Token- and sentence-level perturbations can reliably elicit such outputs (Dong et al., 2021a;b; Zhao et al., 2022; Ye et al., 2022; Huang & Chang, 2021), and audits of ChatGPT have revealed substantial susceptibility (Wang et al., 2023). Beyond input-level perturbations, black-box jailbreak frameworks have been proposed to automate the discovery of exploit templates (Yu et al., 2023). Moreover, even a small number of harmful instruction–response exemplars can serve as few-shot triggers that compromise model alignment (Qi et al., 2024). Zeng et al. (2024) further applies a persuasion taxonomy to generate adversarial prompts that effectively jailbreak LLMs. These findings underscore the need for principled robustness and defense analyses.

**LLM for TQA.**    Table-based reasoning has advanced significantly with LLMs and their emergent reasoning capabilities (Deng et al., 2024; Liu et al., 2024; Su et al., 2024; Wang et al., 2024). To utilize their capability, table contents are typically linearized into a textual sequence and included in the prompt as part of the model input. Rajkumar et al. (2022) use in-context examples for SQL generation, while Cheng et al. (2023) prompt LLMs to produce executable programs via SQL APIs. Lin et al. (2023) extracts sub-tables containing the most relevant information, and Ye et al. (2023b) enhances end-to-end reasoning by decomposing table contexts and questions using few-shot prompting, while Nguyen et al. (2025) decomposes the query into atomic steps for interpretable answers.

**Robustness Evaluation for TQA.**    Several studies have highlighted the robustness limitations of TQA systems. For instance, Chen et al. (2023) introduces permutation-invariant table representations, while Wang & Sun (2022) extends this approach to multi-table scenarios. Bhandari et al. (2025) investigates the robustness of LLMs for TQA under domain shift, and Yang et al. (2022) demonstrates that row and column order can significantly influence model predictions. In related work, Zong et al. (2023) prompt LLMs to generate adversarial examples to improve model robustness during training. However, these investigations are largely empirical—typically involving random permutations of table structures followed by performance observation—thus failing to systematically characterize the worst-case effects of structural perturbations under in-context LLM inference.

Table 1: Alignment scores by LLM-as-judge based on Gemini 2.5 between the responses of different LLMs and the ground truth response, on WTQ dataset, under (i) Vanilla: vanilla input, (ii) Rand Perm: randomly permuted table as input, and (iii) ATP Attack: table permuted by ATP attack as input. The higher score, the better alignment between the model response with the ground truth.

| LLMs | WTQ Dataset | | | | | |
| | Training Set | | | Evaluation Set | | |
| | Vanilla | Rand Perm | ATP Attack | Vanilla | Rand Perm | ATP Attack |
|---|---|---|---|---|---|---|
| LLAMA-3.1-8B | 0.26 | 0.19 | **0.13** | 0.26 | 0.17 | **0.13** |
| LLAMA-3.1-8B-INST | 0.43 | 0.29 | **0.21** | 0.47 | 0.31 | **0.22** |
| TABLELLM-8B | 0.29 | 0.20 | **0.14** | 0.33 | 0.23 | **0.16** |
| QWEN2.5-7B-INST | 0.26 | 0.21 | **0.14** | 0.29 | 0.21 | **0.12** |
| QWEN2.5-14B-INST | 0.44 | 0.33 | **0.24** | 0.47 | 0.33 | **0.26** |
| CODELLAMA-7B-INST | 0.20 | 0.15 | **0.08** | 0.18 | 0.16 | **0.09** |
| DS-R1-DIST-LLAMA-8B | 0.26 | 0.18 | **0.12** | 0.23 | 0.15 | **0.09** |
| DS-R1-DIST-QWEN-7B | 0.16 | 0.11 | **0.07** | 0.14 | 0.12 | **0.06** |

Table 2: Alignment scores of the responses of different LLMs on TATQA dataset under attack.

| LLMs | TATQA Dataset | | | | | |
| | Training Set | | | Evaluation Set | | |
| | Vanilla | Rand Perm | ATP Attack | Vanilla | Rand Perm | ATP Attack |
|---|---|---|---|---|---|---|
| LLAMA-3.1-8B | 0.23 | 0.15 | **0.08** | 0.28 | 0.15 | **0.11** |
| LLAMA-3.1-8B-INST | 0.50 | 0.28 | **0.19** | 0.49 | 0.28 | **0.20** |
| TABLELLM-8B | 0.30 | 0.17 | **0.11** | 0.25 | 0.18 | **0.12** |
| QWEN2.5-7B-INST | 0.28 | 0.15 | **0.11** | 0.28 | 0.20 | **0.13** |
| QWEN2.5-14B-INST | 0.45 | 0.24 | **0.17** | 0.47 | 0.30 | **0.24** |
| CODELLAMA-7B-INST | 0.08 | 0.06 | **0.03** | 0.07 | 0.05 | **0.03** |
| DS-R1-DIST-LLAMA-8B | 0.28 | 0.17 | **0.12** | 0.27 | 0.17 | **0.12** |
| DS-R1-DIST-QWEN-7B | 0.11 | 0.08 | **0.04** | 0.11 | 0.08 | **0.05** |

## 5 EXPERIMENTS

In this section we focus on answering the two previous mentioned research questions: to what extent are current LLMs robust to random table permutations and the worst case table permutations, respectively? In addition, we also conduct ablation study to investigate the effectiveness of each design of the proposed ATP attack. We begin with the experimental settings.

### 5.1 EXPERIMENTAL SETTINGS

**Datasets.** We focus on TQA tasks to evaluate the robustness of LLMs. Specifically, we use three famous document-embedded TQA datasets, WikiTQ (Pasupat & Liang, 2015), and TATQA (Zhu et al., 2021), and FeTaQA (Nan et al., 2022), following Zhang et al. (2024). For each dataset, 1000 random samples from the training set and all the samples from the evaluation set are used.

**Evaluation Metric.** In the literature of TQA, various evaluation metrics have been considered, such as exact match, BLEU (Papineni et al., 2002), ROUGE-L (Lin, 2004), and LLM-as-judge (Zheng et al., 2023). Similar to Zhang et al. (2024), we also found that the LLM-as-judge works the best in terms of evaluating whether the model response is aligned with the reference response. Therefore, we mainly focus on using LLM-as-judge based on Gemini-2.5 (Comanici et al., 2025). To further support our choice, we conduct human evaluation and calculate the rank correlation between the score by human and the score by each metric. We found that the LLM-as-judge achieves the highest rank correlation (around 0.85) with human rating, and it surpasses the runner-up with a clear margin. More detailed discussion about the metric and the rank correlation result can be found in Section A.2.

**Victim LLMs.** We consider several recent and popular open-source LLMs. These are Llama family (Dubey et al., 2024) including Llama-3.1-8B and Llama-3.1-8B-Instruct, Qwen family (Bai et al., 2023) including Qwen2.5-7B-Instruct and Qwen2.5-14B-Instruct, DeepSeek family (Guo et al., 2025) including DeepSeek-R1-Distill-Llama-8B and DeepSeek-R1-Distill-Qwen-7B (due to memory limit

Table 3: Alignment scores of the responses of different LLMs on FeTaQA dataset under attack.

| LLMs | FeTaQA Dataset | | | | | |
| | Training Set | | | Evaluation Set | | |
| | Vanilla | Rand Perm | ATP Attack | Vanilla | Rand Perm | ATP Attack |
|---|---|---|---|---|---|---|
| LLAMA-3.1-8B | 0.38 | 0.28 | **0.21** | 0.36 | 0.25 | **0.20** |
| LLAMA-3.1-8B-INST | 0.50 | 0.41 | **0.30** | 0.48 | 0.37 | **0.31** |
| TABLELLM-8B | 0.29 | 0.24 | **0.16** | 0.34 | 0.27 | **0.20** |
| QWEN2.5-7B-INST | 0.30 | 0.23 | **0.17** | 0.30 | 0.25 | **0.14** |
| QWEN2.5-14B-INST | 0.48 | 0.39 | **0.28** | 0.47 | 0.40 | **0.29** |
| CODELLAMA-7B-INST | 0.20 | 0.15 | **0.09** | 0.23 | 0.19 | **0.13** |
| DS-R1-DIST-LLAMA-8B | 0.29 | 0.22 | **0.13** | 0.28 | 0.21 | **0.13** |
| DS-R1-DIST-QWEN-7B | 0.14 | 0.11 | **0.05** | 0.14 | 0.11 | **0.06** |

Table 4: Ablation study on the hyper-parameters $\lambda_1$ and $\lambda_2$ by the alignment score by LLM-as-judge. Generally $\lambda_1 = \lambda_2 = 10$ performs the best.

| LLMs | WTQ Dataset | | | | |
| | Evaluation Set Against ATP Attack with different values of $\lambda_1, \lambda_2$ | | | | |
| | $\lambda_1, \lambda_2 = 0.0$ | $\lambda_1, \lambda_2 = 0.1$ | $\lambda_1, \lambda_2 = 1$ | $\lambda_1, \lambda_2 = 10$ | $\lambda_1, \lambda_2 = 20$ |
|---|---|---|---|---|---|
| LLAMA-3.1-8B-INST | 0.24 | 0.24 | 0.23 | **0.22** | 0.24 |
| QWEN2.5-7B-INST | 0.15 | 0.14 | 0.13 | **0.12** | 0.14 |

| LLMs | TATQA Dataset | | | | |
| | Evaluation Set Against ATP Attack with different values of $\lambda_1, \lambda_2$ | | | | |
| | $\lambda_1, \lambda_2 = 0.0$ | $\lambda_1, \lambda_2 = 0.1$ | $\lambda_1, \lambda_2 = 1$ | $\lambda_1, \lambda_2 = 10$ | $\lambda_1, \lambda_2 = 20$ |
|---|---|---|---|---|---|
| LLAMA-3.1-8B-INST | 0.22 | 0.22 | **0.19** | 0.20 | 0.22 |
| QWEN2.5-7B-INST | 0.15 | 0.15 | 0.14 | **0.13** | 0.15 |

| LLMs | FeTaQA Dataset | | | | |
| | Evaluation Set Against ATP Attack with different values of $\lambda_1, \lambda_2$ | | | | |
| | $\lambda_1, \lambda_2 = 0.0$ | $\lambda_1, \lambda_2 = 0.1$ | $\lambda_1, \lambda_2 = 1$ | $\lambda_1, \lambda_2 = 10$ | $\lambda_1, \lambda_2 = 20$ |
|---|---|---|---|---|---|
| LLAMA-3.1-8B-INST | 0.33 | 0.33 | 0.32 | **0.31** | 0.33 |
| QWEN2.5-7B-INST | 0.18 | 0.16 | 0.15 | **0.14** | 0.17 |

we consider distilled models for DeepSeek family), CodeLlama-7B-Instruct (Roziere et al., 2023), and TableLLM-8B that is specifically finetuned for TQA tasks Zhang et al. (2024).

## 5.2 MAIN RESULTS

**Robustness of Victim LLMs Against Random Permutations.** The result for WTQ, TATQA, and FeTaQA datasets can be found in Table 1, Table 2, and Table 3 respectively. As we can see, none of the considered LLMs is robust to even random permutations. For example, on WTQ training set, the best performing LLM given vanilla input is Qwen2.5-14B-Instruct, which achieves an average of 0.44 alignment score. When attacked by random permutations, the performance of Qwen2.5-14B-Instruct drops from 0.44 to 0.33. As for TATQA training set, the best performing LLM given vanilla input is Llama-3.1-8B-Instruct, which achieves 0.50 alignment score. The performance decreases to 0.28 given random permutations. On FeTaQA training set, the best one given vanilla input is also Llama-3.1-8B-Instruct. Yet, this performance decreases from 0.50 to 0.41 given randomly permuted tables.

**Robustness of Victim LLMs Against ATP Attack.** Here we are interest in the robustness of modern LLMs against the worst case table permutations. As the exact worst-case solution requires solving a combinatorial optimization in Equation (4) that is NP-hard, we use our proposed ATP attack method (as in Equation (14)) to approximates the worst case scenario. The result for WTQ, TATQA, and FeTaQA datasets is shown in Table 1, Table 2, and Table 3 respectively. We found that, current LLMs are very vulnerable to these adversarially permuted input tables. For example, on WTQ evaluation set, the best performing LLM against ATP attack is Qwen2.5-14B-Instruct, which achieves an average of 0.26 alignment score under ATP attack. However, it still suffers from a significant drop of 0.21 as its performance given vanilla data is 0.47.

These two main empirical results reveal a fundamental defect in the current LLMs when handling tabular data: they are not robust given randomly permuted tables, and are very vulnerable against adversarially permuted input.

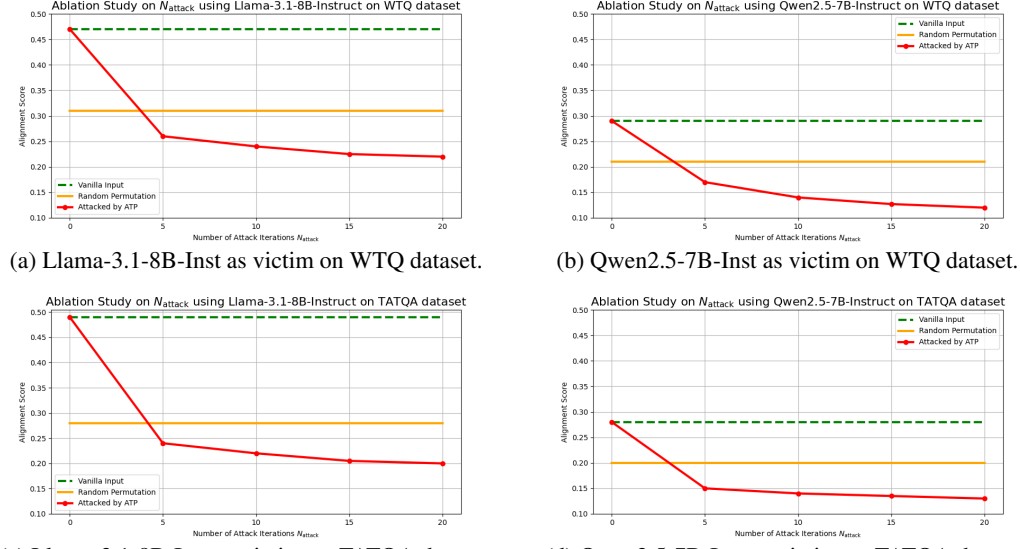

(a) Llama-3.1-8B-Inst as victim on WTQ dataset.   (b) Qwen2.5-7B-Inst as victim on WTQ dataset.

(c) Llama-3.1-8B-Inst as victim on TATQA dataset.  (d) Qwen2.5-7B-Inst as victim on TATQA dataset.

Figure 3: Influence of different values of $N_{\text{attack}}$, the number of attack iterations, on the power of ATP.

### 5.3 ABLATION STUDY

Here we conduct ablation studies to investigate (i) the effectiveness of the entropy regularization term, and (ii) how the number of attack iterations $N_{\text{attack}}$ influence the attack performance. The ablation study on the entropy term can be found in Table 4, where having $\lambda_1 = \lambda_2 = 10$ consistently achieves better attack performance than $\lambda_1 = \lambda_2 = 0$ (i.e., without the entropy regularization terms). For example, on FeTaQa evaluation set, without the entropy term the ATP attack can only decrease the performance of Qwen2.5-7B-Inst from 0.30 to 0.18, while with $\lambda_1 = \lambda_2 = 10$, ATP attack can further degrade the performance of Qwen2.5-7B-Inst to 0.14. At the same time, we notice that a too big $\lambda_1, \lambda_2$ cannot achieve the best attack power either. The reason lies in that with too big $\lambda_1, \lambda_2$, the optimization focuses too much on "hardening" the soft permutations, instead of fooling the victim model. As shown in the table, $\lambda_1 = \lambda_2 = 10$ generally works the best, and thus it is used for ATP attack in our main result in Section 5.2.

As for $N_{\text{attack}}$, the result showing the influence of $N_{\text{attack}}$ on the power of the ATP attack is in Figure 3. We found that ATP with 5 iterations are enough to fool LLMs better than random permutations. For example, as shown in (a) Figure 3, ATP with $N_{\text{attack}} = 5$ degrades the performance of Llama-3.1-8B-Inst from 0.47 to 0.26 while random permutation can only decrease it to 0.31. At the same time, a bigger $N_{\text{attack}}$ generally results in better attack power, which starts to converge around $N_{\text{attack}} = 20$. Thus $N_{\text{attack}} = 20$ is used in our main result in Section 5.2.

### 5.4 RUNTIME ANALYSIS AND DISCUSSIONS ON ATTACKING CLOSED SOURCE LLMS

The computation complexity of ATP attack largely depends on the hyper-parameter $N_{\text{attack}}$. In our implementation with a single A100 GPU, it takes around 10 seconds for ATP with $N_{\text{attack}} = 20$ to attack a data point. We can also trade some attack power off by decreasing $N_{\text{attack}}$ to 5, which can still fool modern LLMs to a considerable extent and takes only around 3 seconds per sample.

We note that our ATP attack is a gradient-based attack method and requires the access of gradients of a victim model. Thus, for those closed source LLMs such as Gemini and ChatGPT, ATP cannot be directly deployed. However, as long as we can query the model, methods such as zero-order optimization can be employed to enable ATP to work in such a black-box attack scenario. We leave this direction for future explorations.

## 6 CONCLUSION

In this work, we demonstrate the susceptibility of modern LLMs to row and column permutations and then formalize this permutation sensitivity for TQA. Building on this, we introduce Adversarial Table Permutation (ATP), attack that finds worst-case permutations to fool a victim model. We show that ATP consistently uncovers worst-case permutations that significantly degrade the performance of a various modern LLMs.

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

# A APPENDIX

## A.1 USAGE OF LLM

We employed LLMs as an auxiliary tool to polish the grammar and clarity of our manuscript. This usage was limited to improving readability and presentation; all conceptual contributions, analyses, and interpretations were developed by the authors.

## A.2 METRIC USED FOR ALIGNMENT SCORES

We further illustrate the effect of the ATP attack by contrasting model outputs generated from the original table input with those obtained from its adversarially permuted counterpart, thereby validating the effectiveness of our evaluation metrics. For each instance, we generate outputs under both conditions and compute example-level alignment scores with respect to the reference answer. We report three metrics to assess the similarity between generated and reference outputs:

- **ROUGE-L (Lin, 2004)**: lexical similarity via longest common subsequence between generated and reference answers.

- **LLM-as-judge** (Zheng et al., 2023): a held-out LLM (Gemini 2.5 (Comanici et al., 2025)) scores the semantic similarity between the model's output and the ground truth.

- **Human annotation**: human raters assess semantic similarity relative to the ground truth answer.

**Method.** We uniformly sample TQA instances in the dataset. For each instance, we construct two prompts (original vs. ATP-permuted table), prompt the model to generate outputs, and score them with ROUGE-L, the LLM-as-judge, and human raters. To assess agreement among metrics, we compute Spearman's rank correlation over example-level scores for each metric pair. The resulting correlations are summarized in Figure 4.

Based on these results, we observe that the LLM-as-judge metric exhibits strong alignment with human judgments, whereas ROUGE-L fails to capture such alignment. This finding supports the effectiveness of our LLM-as-judge as an evaluation metric for measuring alignment score between LLM-generated responses and ground-truth answers. In contrast, metrics like ROUGE-L, which rely on lexical overlap, are limited to token-level similarity and thus fall short in evaluating semantic equivalence, especially in tasks involving long-form reasoning in TQA task. We further illustrate this limitation with an example in which the adversarially perturbed (ATP-attacked) table input flips the model's originally correct prediction(see Section A.3). Although the perturbed output achieves a higher ROUGE-L score due to surface-level overlap, it no longer preserves the correct semantics. Conversely, the LLM-as-judge metric successfully detects this change in prediction, validating its suitability for robust evaluation in such scenarios.

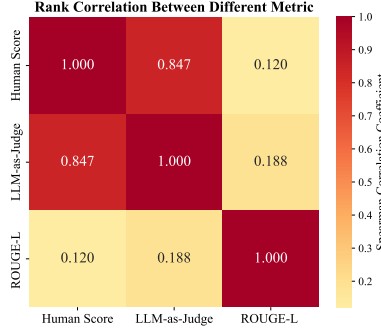

Figure 4: Spearman's Rank Correlation between different metrics. The LLM-as-judge method showed strong alignment with the human rater.

We also provide an illustrative example below to show how ATP permutes the input table to fool a victim LLM. In the example we can also see that the LLM-as-judge works better than Rouge-L. More specifically, given original input, the model gives a correct response of "Desmond Trufant" while with permuted input, the model outputs an incorrect answer of "Robert Alford", Note that the ground truth is "Desmond Trufant". Rouge-L gives a score of 0.37 for the correct response but gives 0.39 to the incorrect response, which is not ideal. As a contrast, LLM-as-judge gives 0.8 to the correct one and gives 0.1 to the incorrect response, which aligns well with human rating.

## A.3 AN ILLUSTRATIVE EXAMPLE FOR ATP ATTACKED INPUT

---

**Example of ATP Attack Result (by Qwen2.5-14B-Instruct)**

**Instruction.** Offer a thorough and accurate solution that directly addresses the Question outlined in the [Question].

[Table]
```
Round|Selection|Player|Position|College|
1|22|Desmond Trufant|CB|Washington|
2|60|Robert Alford|CB|Southeastern Louisiana|
4|127|Malliciah Goodman|DE|Clemson|
4|133|Levine Toilolo|TE|Stanford|
5|153|Stansly Maponga|DE|TCU|
7|243|Kemal Ishmael|S|Central Florida|
7|244|Zeke Motta|S|Notre Dame|
7|249|Sean Renfree|QB|Duke|
```

[Table Description]
This table summarizes an NFL draft class drawn from college programs. For each pick, it lists the round, overall selection number, player name, position, and college.

[Question]
Who is the top listed player?

- - - - - - - - - - - - - - - - - - - - - - - - - - - - - - - - - - - - - - - - - - - - - - - - - -

**Original Answer** (Rouge-L: 0.37, LLM-as-judge Score: 0.8)

[Solution]Among the athletes listed in the draft class table, the first corresponds to **Desmond Trufant**, who was selected in the opening round of the draft, holding the 22nd overall position.

- - - - - - - - - - - - - - - - - - - - - - - - - - - - - - - - - - - - - - - - - - - - - - - - - -

**Attacked Instruction.** Offer a thorough and accurate solution that directly addresses the Question outlined in the [Question].

[Table] (Permuted by ATP Attack)
```
College|Round|Player|Selection|Position|
Clemson|4|Malliciah Goodman|127|DE|
Stanford|4|Levine Toilolo|133|TE|
Duke|7|Sean Renfree|249|QB|
Central Florida|7|Kemal Ishmael|243|S|
Washington|1|Desmond Trufant|22|CB|
Notre Dame|7|Zeke Motta|244|S|
TCU|5|Stansly Maponga|153|DE|
Southeastern Louisiana|2|Robert Alford|60|CB|
```

[Table Description]
This table summarizes an NFL draft class drawn from college programs. For each pick, it lists the round, overall selection number, player name, position, and college.
[Question]
Who is the top listed player?

- - - - - - - - - - - - - - - - - - - - - - - - - - - - - - - - - - - - - - - - - - - - - - - - - -

**Attacked Answer** (Rouge-L: 0.39, LLM-as-judge Score: 0.1)

[Solution]The top-listed player in your table is **Robert Alford** from Southeastern Louisiana, where he played as a Cornerback. He was selected the 2nd pick 60th in Round 2.

- - - - - - - - - - - - - - - - - - - - - - - - - - - - - - - - - - - - - - - - - - - - - - - - - -

[Ground Truth]
The top listed player is **Desmond Trufant**, who was selected in Round 1 with the 22nd pick. He played as a cornerback and attended the University of Washington.

---

