# OpenReview forum: "The Power of Order: Fooling LLMs with Adversarial Table Permutation"
_ICLR.cc/2026/Conference — Submitted to ICLR 2026_

### Official Review · Reviewer_T71A · 2025-10-27

**Soundness:** 2
**Presentation:** 3
**Contribution:** 3
**Rating:** 6
**Confidence:** 4

**Summary:**

This paper introduces a permutation-based attack on LLMs. The core premise is that current LLMs are sensitive to layout over content. The authors then investigate this by defining an attack space consisting of row and column permutations that keep the semantics of the table preserved. Then, they introduce a gradient based search method to find a worse case permutation for given question answer table triplets (Adversarial Table Permutation). ATP works by relaxing permutations into soft permutations, but projecting hard permutations for position ids. It is evaluated on several open source LLMs at the 7-14B range, and uses Gemini 2.5 as the judge (0.85 spearman with humans). Results indicate that alignment drops with permutations on WTQ, TATQA, FeTaQA.

**Strengths:**

1. Clear formulation of attack strategy that’s reproducible. The authors provide clear details on legal permutations, and formulations regarding robustness as invariance to permutations.
2. ATP is a novel attack method using differentiable relaxation of a combinatorial search over permutations via Sinkhorn, sharpening them, then project back to valid permutations for discrete inputs.
3. Good coverage of LLMs with Llama, Deepseek, etc on 3 table datasets.
4. Analysis clearly highlights central claim: permutations influence accuracy, and ATP is strictly worse.
5. Decent grounding of LLM judge vs human agreement and against heuristic to justify choice.
6. Practical insight from a robustness perspective on how a semantic no-op can lead to accuracy issues with production models. Very much related to literature in context rot and forgetfulness of LLMs.

**Weaknesses:**

1. ATP is reliant on gradients from open source LLMs, which is not a reasonable assumption. Authors should expand their test analysis on closed source models after training, or offer discussion on how ATP can be used in conjunction with closed source API systems.
2. Realism of the attack is not well justified, tables are often ordered or imply chronology. It’s not clear how often scrambled tables would appear in reality. Additional discussion should relate to this position.
3. Causality of robustness claims is correlational. The lack of analysis is lacking on where and what types of permutations cause the steepest drops. Do the first row and column matter more than others? Is there any insight if permutations were restricted to subsets only?
4. Lack of experiments for defenses to permutations attack. Could this attack be thwarted by a simple sort operation by LLM predicting the canonical order of columns before TQA, or rewriting the question for explicit row/column values?
5. The drop in alignment is not clear in understanding. Would 0.26 still be usable but in degraded state or completely useless? Expanding the interoperability of the scale here would be beneficial, and more quantitative metrics on where, in aggregate, ATP flips answers from correct to incorrect would strengthen the claim.

**Questions:**

1. Can the authors elaborate on the circumstances where an attacker would have access to a victim model for gradients? How can ATP generalize from open source to closed source systems?
2. Have you evaluated a constrained version of ATP that produces plausible yet scrambled tables? What if order was visually reasonable, or rows shuffled within groups or at certain locations (head rows vs tail rows)?
3. What defensive mechanism could be used to counter permutation based attacks? Could reasoning about a canonical order mitigate the decrease in robustness?
4. What percent of the questions experienced factual flips like your example? What percent of questions flipped being semantically aligned to misaligned?
5. How can you prove that ATP is finding the worse case permutation rather than approximating it? Did you grid search across a small subset for all possible permutations to find the worst?

---

### Official Review · Reviewer_qjQs · 2025-10-29

**Soundness:** 2
**Presentation:** 1
**Contribution:** 1
**Rating:** 2
**Confidence:** 4

**Summary:**

This paper studies a vulnerability in Large Language Models (LLMs) for Table Question Answering (TQA): their sensitivity to semantically-invariant row and column permutations. The authors propose Adversarial Table Permutation (ATP), a gradient-based attack that identifies worst-case permutations by relaxing the discrete permutation space into a continuous one using doubly stochastic matrices and entropy regularization. Experiments on multiple benchmarks (WTQ, TATQA, FeTaQA) and models show that both random and adversarial permutations significantly degrade model performance.

**Strengths:**

1. The paper applies an existing technical method—adversarial permutation optimization via the Sinkhorn algorithm—to the table setting, where permutation sensitivity is a well-known challenge.

2. Experiments demonstrate that the method is effective across different datasets and models.

**Weaknesses:**

This paper bears significant similarity to a ICLR 2025 publication [1] which is not cited. Both papers use the same technique: relaxing permutation matrices to doubly stochastic matrices, applying Sinkhorn algorithm, and using entropy regularization. The framework appears identical, with the main difference being the application scenario (table permutation vs. ICL demonstration permutation).

Given that [1] was already publicly available on OpenReview when this paper was submitted, the absence of citation is deeply concerning. The authors should clarify the relationship between these works and properly acknowledge prior art. Without such clarification, it is difficult to assess the novelty of this submission.

[1] PEARL: Towards Permutation-Resilient LLMs.

**Questions:**

No

---

### Official Review · Reviewer_UhhG · 2025-10-31

**Soundness:** 3
**Presentation:** 2
**Contribution:** 2
**Rating:** 2
**Confidence:** 4

**Summary:**

This paper identifies and systematically investigates a critical yet overlooked vulnerability in Large Language Models (LLMs): their lack of robustness to semantically-invariant permutations in tabular data. The authors convincingly demonstrate that merely rearranging the rows and columns of a table can lead to inconsistent or incorrect outputs from state-of-the-art LLMs.

**Strengths:**

1. Identifies a Fundamental Flaw: Exposes a critical lack of structural robustness in LLMs applied to tabular data, moving beyond content-based vulnerabilities.

2. Proposes a Novel Attack: Introduces ATP as an effective and efficient method for probing this specific vulnerability.

**Weaknesses:**

I have several concerns regarding the experimental section of this paper:

1. Insufficient baseline comparison: The experiments only include a simple comparison against random permutation, lacking comparisons with established or state-of-the-art baseline methods. This makes it difficult to objectively evaluate the relative performance of the proposed approach.

2. Limited evaluation metrics: The paper focuses solely on attack-oriented metrics (e.g., attack success rate) while neglecting other crucial dimensions, such as the number of model queries required or the computational cost required to mount the attack, which are essential for assessing practical efficiency.

3. Limited model scale for experiments: The validation is conducted only on 7B and 8B parameter LLMs. It is recommended to include supplementary experiments on larger-scale LLMs (e.g., 70B parameters or more) to demonstrate the generality of the identified vulnerability and the scalability of the method.

**Questions:**

1. Figure quality: The authors are advised to use vector graphics (e.g., PDF or EPS) instead of the current raster images. The existing figures exhibit significant pixelation when zoomed in, compromising their clarity and professional presentation.

2. Inaccurate terminology: The statement on line 198 should be correctly categorized as a "Lemma." Its rigor and generality do not meet the standard for a "Theorem," and the terminology should be revised to adhere to academic conventions.

---

### Official Review · Reviewer_35Wc · 2025-10-31

**Soundness:** 3
**Presentation:** 2
**Contribution:** 3
**Rating:** 4
**Confidence:** 4

**Summary:**

This paper examines the phenomenon on adversarially prompting LLMs for Tabular question answering (TQA) and show that even naive permutations make LLMs susceptible and results in unreliable responses. They propose a gradient-based framework, Adversarial Table Permutations (ATP) to come up with best permutation possible to degrade the performance of LLMs on TQA tasks.

**Strengths:**

- Well motivated, clearly written and experiments demonstrate the claims.
- I like the idea of even small permutations affecting the model’s performance. Was surprised that there are no previous methods that studied this phenomenon (refer “relevant baselines” comment in weaknesses).

**Weaknesses:**

**Experiment on relevant baselines**: Aren’t there any optimization-based baselines to compare against?

**Additional baselines**: Also, please augment with the previous works the authors have cited [1, 2] for random permutations to make sure the comparison is thoroughly done.

**Experiment 1**: It would be interesting to understand the effect of ATP/random perturbations on tabular synthetic data [3, 4]. Questions such as whether LLMs are vulnerable to attacks using synthetic data and if so whether it’s more/less compared with real data (WTQ etc;) and to what extent (maybe random perturbations are enough if it’s synthetic data as it might have something inherently adversarial!) can be answered to an extent. Basically the success of ATP depends on the time spent on the optimization performed and the actual data used. For example, both [3, 4] use a form of randomly shuffling data to train the models for invariant feature, so it will be interesting to understand the consequences of using synthetic data generated from such models.

**Experiment 2**: Can the authors comment on scenarios when there’s data dependency in the rows/columns? Let’s say activity level data such as in healthcare, financial domains. You can consider an experiment around some datasets and report any interesting findings:
- For eg: If a table has the following structure, we know that the future price (somehow) depends on past price and we can’t really look at each row as an independent point. There are cases when columns are dependent as well.
| Stock | Date        | Price |
|--------|-------------|-------|
| A       |  1 Jan 2025  | 100   |
| A       | 1 Feb 2025  | 200   |

L102-104: I see there’s iid assumption, but wanted to hear the authors thoughts on these scenarios and a potential experiment as it’s a practical scenario of using LLMs for TQA with interdependent rows/columns potentially breaking iid assumption.

**Experiment 3**: An experiment on model sizes of same family vs ATP (attack iterations $N_\{attack}$ let’s say) will be helpful. Example can be: Pick 1B, 3B, 8B, 13B etc; and see how easy/difficult are the models across scale with respect to the attack. This supplements L22,23 of abstract.

Feel free to make design choices depending on compute (i.e by picking latest methods) and if any experiment is not possible, please describe the reasoning in detail.

1. Tableformer - https://arxiv.org/abs/2203.00274
2. Transtab - https://arxiv.org/abs/2205.09328
3. Tabby: Tabular Data Synthesis with Language Models: https://arxiv.org/abs/2503.02152
4. Language Models are Realistic Tabular Data Generators: https://arxiv.org/abs/2210.06280

**Questions:**

Some feedback to improve the presentation.

- While the authors has discussed the effect of ATP/gradient-based methods on closed-source models briefly in the conclusion, I suggest to expand the discussion on what will enable such an attack for closed source models.
- L140-148: Consider adding more details in a section at appendix as why it’s NP-hard and how n=m=8 got 1.6x10^9 combinations (8! X 8!).
- Please add a section in Appendix describing the datasets in detail.
- WikiTQ vs WTQ: Maintain only one notion in the paper.
- L206, L223, L247: Check “ vs ``. L206: “transform them”
- Is there a motivation to pick 1000 random samples from training set and all of evaluation set? How will that affect the results as the training set/evaluation set are technically the same for these models. Consider elaborating more in the paper if needed.
- [A2]. Can you elaborate more on the “Human annotation: human raters assess semantic similarity relative to the ground truth answer.” i.e how did the semantic similarity measured by humans.
- [A2] Consider adding the actual values for LLM-as-a-judge, human ratings supplementing Fig.4 similar to Table 3. It will be helpful for reproducibility purposes in future as LLM-as-a-judge values are not always reproducible (the end-points for closed model changes), so the other metrics will come in-handy. Also, consider open-sourcing the codebase.
- I am not sure but \citet vs \citep might have an affect in the way I see the citations. Eg: L363, 364 - Some are present in () while some are in free flow text.

---

### Meta-Review · Area_Chair_iiC3 · 2025-12-26

**Summary:**

This paper studies the robustness of large language models for table question answering to semantically invariant row and column permutations. It proposes a gradient-based attack, Adversarial Table Permutation (ATP), which finds worst-case permutations by relaxing the problem into a continuous space using doubly stochastic matrices, the Sinkhorn algorithm, and entropy regularization. Reviewers acknowledged the problem's importance and the paper's clear demonstration of the vulnerability. However, they also raised significant concerns regarding the method's novelty relative to closely related work, the limited set of baselines and evaluation metrics (which omit query and computation costs), the small scale of the models tested, a reliance on gradient access, a shallow analysis of which permutations are most harmful, the absence of proposed defenses, and several presentation issues.

**Reviewer Concerns:**

As no author rebuttal was provided, all major reviewer concerns remain outstanding. These include the need to clarify novelty and positioning against prior art, the call for stronger baselines and cost-aware evaluation, the request to test on larger or black-box models, the desire for a deeper analysis of failure modes and potential defenses, and the need to fix various presentation errors.

**Reviewer Scores:**

After discussion, reviewer 35Wc would likely maintain their score of 2 (reject), as would UhhG and qjQs. Reviewer T71A, who was initially more positive with a 6, would likely soften their score after considering the community's concerns about novelty and the limited baselines.

---

### Decision · Program_Chairs · 2026-01-26

Reject